# "Behavioural and metabolic risk factors of cardiovascular diseases among post-menopausal women: A cross-sectional study in Itahari sub-metropolitan city of Nepal"

**Suraksha Khatri**[1]*, **Deepak Kumar Yadav**[2], **Anup Ghimire**[2], **Dharanidhar Baral**[2], **Birendra Kumar Yadav**[2], **Paras Kumar Pokharel**[2]

1 School of Public Health and Community Medicine, B.P. Koirala Institute of Health Sciences, Dharan, Nepal,
2 Faculty of School of Public Health and Community Medicine, B.P. Koirala Institute of Health Sciences, Dharan, Nepal

* khatrisuraksha65@gmail.com

## Abstract

### Background and objectives

Cardiovascular diseases (CVDs) are main cause of mortality and morbidity among women globally. Menopause, aging and different factors in collaboration leads to increased risk for CVDs among postmenopausal women but study on risk factors of CVDs among postmenopausal women is limited in Nepal. Therefore, we aimed to assess prevalence of behavioural and metabolic risk factors of cardiovascular disease among post-menopausal women.

### Materials & methods

Community based cross-sectional study was carried among 390 post-menopausal women of the Itahari sub-metropolitan city. Multistage simple random sampling technique was used to collect data. Modified semi structured questionnaire STEP I, STEP II and STEP III was used as data collection tool. SPSS version 11.5 used for data analysis. Chi-square test and logistic regressions were performed at 95% confidence interval to find predictors of raised blood pressure and overweight/obesity.

### Results

The mean age of respondents was 63.37 years. The prevalence of current smokers, current alcohol use, less than 5 servings of fruits and vegetables per day, low level of physical activity, overweight/obesity and raised blood pressure was 9.2%, 18.2%, 86.4%, 52.3%, 56.2% and 44.4% respectively. On multivariate analysis strong predictors of overweight obesity were age ≤59 years (AOR: 3.21, CI 1.49–6.89), poor economic status (AOR: 1.764, CI: 1.120–2.779) and low physical activity (AOR: 2.132, CI: 1.350–3.370). Predictors of raised blood pressure were age ≥75 years (AOR: 2.04, CI: 1.01–4.11), unmarried and widow (AOR: 1.815, CI: 1.108–2.971), higher economic status (AOR: 1.752, CI: 1.119–2.740) and smoking (AOR: 4.109, CI: 1.737–9.718).

**Data Availability Statement:** All relevant data are within the paper.

**Funding:** The funders had no role in study design, data collection and analysis, decision to publish, or preparation of the manuscript.

**Competing interests:** The authors have declared that no competing interests exist.

## Conclusion

Prevalence of behavioural and metabolic risk factors among postmenopausal women in Itahari sub-metropolitan city were alarmingly high. This calls for an early need of intervention and policies at local, provincial and institutional level to address risk factors of CVDs.

## Introduction

Cardiovascular diseases (CVDs) are number one leading cause of death worldwide and accounts for 32% of all deaths globally. Three–fourth of CVD deaths occurs in low–and middle-income countries and 85% of cardiovascular disease deaths are due to heart attacks and strokes [1]. In south Asia, Nepal ranked third in prevalence, and the CVD related Disability Adjusted Life years rates also higher in Nepal than the global average [2]. Among noncommunicable diseases (NCDs) heart disease is the number one killer disease at the global level (2000 to 2019) and it is taking more lives than ever before. More than 15 million deaths from NCDs occurs between the ages of 30 and 69 year [3].

Menopause is defined as the cessation of menstruation for 12 months in a woman over the age of 45 years and occurs at a median age of 52 years. Sometimes, menopause occurs earlier, due to diseases, genetic factors, or surgery. The largest health threat to women over age 50 is cardiovascular disease (CVD). In women age 45–49 years, the incidence of CVD is 3 times lower than men of matched age [4, 5]. The incidence and prevalence of coronary heart disease and atherosclerotic diseases are higher in postmenopausal than in premenopausal women [6]. Before menopause the incidence of myocardial infarction in women lower than in men, incidence increases dramatically after menopause [7].

Globally cardiovascular disease is the number one cause of mortality for women and was responsible for 35% of total deaths in women in 2019. However cardiovascular disease in women remains understudied [8]. Estrogen level declines after menopause and CVD risk increases [9]. Usually women experiences first episode of CVD after menopause and this period is landmark for impending CVDs [10]. A community-based case control study from Nepal showed that total cholesterol, low density lipoprotein, high density lipoprotein and triglycerides were highly increased in postmenopausal women compared to pre-menopausal women [11].

Many studies are carried out on NCDs and CVDs but most of the studies focused on general adult population, very few study are conducted among postmenopausal women. With increase in life expectancy of females in Nepal, life spent after postmenopausal period has also increased. Therefore this study is conducted to bridge the gap in the field of study on CVDs among postmenopausal women.

## Methodology

### Study design and setting

A Community based cross-sectional analytical study was carried out among post-menopausal women of the Itahari sub-metropolitan city of Nepal. Itahari sub metropolitan city is business hub of eastern Nepal. It is the second most populous city in Koshi Province after Biratnagar. Furthermore, Itahari is rapidly growing city of Koshi province facing lifestyle changes, nutritional transition and epidemiological transition following westernization and urbanization.

## Study population, sample size and sampling technique

Postmenopausal women residing in Itahari sub-metropolitan city for at least one year were study population of our study. Those who were unable to respond due to cognitive dysfunction, severely ill and with severe mental disorders were excluded from the study.

The sample size is estimated, based on the study by Rupal Dosi et al. on cardiovascular disease and menopause, where the prevalence of hypertension among postmenopausal women was 52% [12]. Formula for estimating the one sample proportion ($n = Z^2pq/e^2$) with margin of error 10% of prevalence, level of significance 5% and non-response rate 10%. Calculated sample size was 390.

Multistage simple random sampling technique was used to select sample. First Itahari sub-metropolitan city was selected as it is the second most populous city in province one. Out of twenty wards in Itahari 4 wards were selected using lottery method than proportionate allocation of sample was done to select samples from each ward. The selected wards were 4, 5, 8 and 19 and study units were 141, 126, 60 and 63 individuals respectively. Bottle was rotated in main junction of ward to select the areas of ward and move towards the direction as directed by bottle neck. Participants were selected from each ward until sample size was reached for respective ward. One respondent was selected from one house. If more than one postmenopausal women present at the time of data collection than one respondent was selected by using lottery method. If no postmenopausal women available than proceeded to adjacent household.

## Data collection tool and technique

Modified semi structured questionnaire based on World Health Organization's validated STEP wise approach to NCD surveillance using STEP I, STEP II and STEP III was used [13]. Face to face interview was carried out, anthropometric and clinical measurement was done. Pretesting was done on 10% of sample size in similar setting. Data collection was carried out by researcher herself.

**Data collection period:-** Data collection was done from 18[th] June 2022 to 20[th] September 2022.

## Behavioural and metabolic risk factors

**Current smoking:** Those who had smoked in past 30 days were considered as current smokers. [14] **Alcohol drinking:Current drinker:-** Those who have drank in past one month or drink on occasional basis. [13] **Heavy episodic drinking:—**Heavy episodic drinking (HED) is defined as consumption of 60 or more grams of pure alcohol (6+ standard drink in most of countries) on at least single occasion in the 30 days prior to the survey. [15] **Low fruits and vegetable intake:** Less than five servings of fruits and vegetables per day was considered as low fruits and vegetables intake. [13] **Salt intake:** Consumption of less than 5 gram of salt per day was considered as normal. [16] **Physical activity:** Physical activity was assessed using the Global Physical Activity Questionnaire (GPAQ). Those who do not meet WHO recommendation on physical activity for health (<600 MET-minutes) were categorised as low physical activity [13]. **Overweight or obesity:** A BMI of $\geq 30$ kg/m$^2$ and between 25.0 kg/m$^2$ and 29.9 kg/m$^2$ were considered as obese and overweight respectively [17]. **Raised blood pressure:** Raised blood pressure was defined as having systolic blood pressure $\geq 140$ mm Hg and /or diastolic blood pressure $\geq 90$ mm Hg during the study, or on medication for raised blood pressure [13]. **Waist hip ratio:** According to WHO cut off points and risk of metabolic complications, a waist hip ratio of more than equal to 0.85 cm in women was consider increased waist hip ratio [18]. **Ethnicity:** It was classified as per health management information system (HMIS) guidelines of Nepal [19]. **Poverty line:** The poverty line was taken in

reference to international poverty line as $1.90 /person/day updated by the World Bank in 2015. If it was <$1.90/person/day, it was categorized as below poverty line. If it was ≥$1.90/person/day, it was categorized as above poverty line [20]. **Lipid profile:** High cholesterol is defined as total cholesterol more than equal to 200mg/dl. Similarly, high triglyceride is defined as triglyceride level more than equal to 150mg/dl [21]. **Postmenopausal women**: Postmenopausal women was identified by the criteria of cessation of menstrual period for more than 12 consecutive months [22].

### Data management and statistical analysis

All questionnaires were filled up during the interview and kept on file. Master chart was prepared at Microsoft Excel and the coding list was prepared manually and the data was entered daily. After completion of data entry, excel file was imported at Statistical Package for Social Sciences (SPSS) 11.5 version for statistical analysis. After every 10 entries, the data were re checked for its accuracy. The descriptive data was presented in frequency, percentage, mean, median, standard deviation and interquartile Range (IQR). Bivariate analysis was done using Chi-square test to find out the association between metabolic risk factors (overweight/obesity and raised blood pressure) of CVDs and other selected variables at 95% confidence interval. Explanatory variables showing p value less than 0.2 in bivariate analysis were entered into logistic model for multivariate analysis. Logistic regression was used to find out predictors of raised blood pressure and overweight/obesity after adjusting all possible associated factors, also to find adjusted odd ratio (AOR) and p-value less than 0.05 at 95% class interval(CI) was considered statistically significant.

### Ethical clearance

The written official ethical consent was obtained from Institutional Review Committee (IRC), BPKIHS, Dharan (Reference number 216/078/079-IRC). Written permission was taken from the Itahari sub-metropolitan city (Reference number 4411/078/079). The purpose of the study and procedure were explained and written informed consent was obtained before proceeding the data collection. The participants were assured of their anonymity, confidentiality and authority to accept or refuse to take part in the study.

## Results

### Socio-demographic characteristics of the participants

The mean (SD) age of participants was 63.37 ± 10.57 years with nearly half (45.1%) of the respondents belonging to age group 60–74 years. Nearly two third (65.6%) of the respondents were married and majority (95.1%) of them follow Hindu religion. More than one third (36.9%) of respondents belong to Janjati. Majority (89.2%) of the participants were illiterate. Most (70.5%) of the participants were home maker. Nearly two third (64.9%) of the respondents belonged to below poverty line. Median family income/month of participants was NRs 25000 with Interquartile range NRs 15000—NRs 40000/-. Most (60.8%) of the respondents had family members ≥ five (Table 1).

### Behavioural and metabolic risk factors of CVDs

**Smoking:** The overall prevalence of current smokers was 9.2%. Among the current smokers 97.2% were daily smokers. Nearly one fourth (23.8%) of respondents were past smokers. Current smokeless tobacco users were 11.3% and all of them were daily users of tobacco. **Alcohol:** Majority of the respondents were lifetime abstainer(79.7%). Current alcohol users 18.2%.

**Table 1. Socio-demographic characteristics of the participants.**

| Characteristics | Category | Number(%) | Characteristics | Category | Number (%) |
|---|---|---|---|---|---|
| Age (in years) | 30–44 | 6(1.5) | Ethnicity | Dalit | 27(6.9) |
| | 45–59 | 142(36.4) | | Janjati | 144(36.9) |
| | 60–74 | 176(45.1) | | Madhesi and Giri | 13(1.5) |
| | 75–89 | 61(15.6) | | Brahmin | 109(27.9) |
| | 90–104 | 5(1.3) | | Chhetri | 97(24.9) |
| **Mean age ± SD (Min–Max)** | | 63.4 ± 10.6 (33–97) | Religion | Hindu | 371(95.1) |
| Marital Status | Married | 256(65.6) | | Buddhist | 13(3.3) |
| | Unmarried | 6(1.5) | | Christian | 3(0.8) |
| | Widow | 128(32.8) | | Muslim and Kirat | 3(0.8) |
| Educational level | Illiterate | 348(89.2) | Economic Status | Below Poverty Line | 253(64.9) |
| | Literate | 42(10.8) | | Above Poverty Line | 137(35.1) |
| Occupation | Home maker | 275(70.5) | Number of family member | <5 | 153(39.2) |
| | Not working | 81(20.8) | | ≥5 | 237(60.8) |
| | Self employed | 34(8.7) | **Median number of family member/IQR** | | 5/ (4–6) |

Fruits and vegetables: Majority of the respondents 86.4% consumed less than 5 serving of fruits and vegetables per day. **Salt:** Most of the respondents (65.6%) consumed $\leq$ 5 gram of salt per day. **Physical activity:** Low level of physical activity was present in 52.3% of the respondents. **Metabolic risk factors:** overweight/obesity present in 56.2% respondents and raised blood pressure in 44.4%. Lipid profile data was obtained from 53 respondents among them 49% had high cholesterol and 58.5% had high triglycerides. Sixty one percent respondents had increased waist hip ratio (Table 2).

## Determinant of metabolic risk factor

Bivariate analysis using chi-square test was done and those factors having p-value less than 0.2 were included for logistic analysis. Bivariate analysis of metabolic risk factors are shown in Table 3.

 **Overweight/obesity:** It was observed that the odds of being overweight/obesity increases by 3.21 times in age group $\leq$59 years than the age group $\geq$ 75 years (AOR:3.21, CI 1.49–6.89). Similarly, odds of being overweight/obesity increases by 1.4 times among age group 60–74 years than age group $\geq$ 75 years (AOR:1.4, CI: 0.71–2.81).It was evident that respondents above poverty line were 1.76 times more likely to be overweight/obesity than those living below poverty line and was statistically significant (AOR: 1.76, CI: 1.12–2.78). Furthermore, it was observed that odds of being overweight/obesity increases by 2.132 times among sampled population doing low physical activity than those doing normal level of physical activity (AOR: 2.13, CI: 1.35–3.37) (Table 4).

 **Raised blood pressure:** Odds of having raised blood pressure was 1.82 times higher among unmarried and widow women than the married women (AOR: 1.82, CI: 1.11–2.97). Similarly, it was observed that respondents belonging to above poverty line were 1.75 times more likely to have raised blood pressure than respondents belonging to below poverty line (AOR: 1.75, CI: 1.12–2.74). Further, smokers were 4 times more likely to have raised blood pressure than nonsmokers (AOR: 4.11, CI: 1.74–9.72) (Table 4).

## Discussion

This community based cross sectional study assessed prevalence of risk factors of CVDs and associated factors of metabolic risk factors of CVDs among postmenopausal women. Study

**Table 2. Behavioural and metabolic risk factors.**

| Variables | Characteristics | Categories | Number(%) |
|---|---|---|---|
| **Bahavioural risk factors of CVDs** | | | |
| **Smoking** | Current smokers | Yes | 36(9.2) |
| | | No | 354(90.8) |
| | Current daily smokers (n = 36) | Yes | 35(97.2) |
| | | No | 1(2.8) |
| | Past smokers | Yes | 93(23.8) |
| | | No | 297(76.2) |
| | Current smokeless tobacco user | Yes | 44(11.3) |
| | | No | 346(88.7) |
| | Current daily smokeless tobacco users | Yes | 44 (100) |
| | | No | 0(0) |
| **Alcohol** | Lifetime abstainers | Yes | 311(79.7) |
| | | No | 79(20.3) |
| | Consumed alcohol in 12 months | Yes | 72(18.5) |
| | | No | 318(81.5) |
| | Heavy episodic drinking (N = 72) | Yes | 1(1.4) |
| | Consumed alcohol in 30 days | Yes | 71(18.2) |
| | | No | 318(81.8 |
| **Fruits and/or vegetables intake** | No. of serving | <5 | 337(86.4) |
| | | ≥5 | 53(13.6) |
| **Salt** | Amount (Gram per day) | ≤5 | 256(65.6) |
| | | >5 | 134(34.4) |
| **Physical activity** | Level of physical activity | Low (<600 MET-minutes) | 204(52.3) |
| | | Moderate (600–2999 Met-minutes) | 184(47.2) |
| | | Vigorous (> 3000 MET—minutes) | 2(0.5) |
| **Metabolic risk factors** | | | |
| **Overweight/Obesity** | >24.9 kg/m2 | Yes | 219(56.2) |
| | | No | 171(43.8) |
| **Raised blood pressure** | systolic ≥140 mm of Hg and or diastolic ≥90 mm of Hg or previously diagnosed hypertension | No | 217(55.6) |
| | | Yes | 173(44.4) |
| **Dyslipidemia** | High Cholesterol | Yes | 26(49) |
| | | No | 27(51) |
| | High triglyceride | Yes | 31(58.5) |
| | | No | 22(41.5) |
| **Waist hip ratio** | Increased waist hip ratio | Yes | 238(61) |
| | | No | 152(49) |

revealed prevalence of current smokers, current alcohol use, less than 5 servings of fruits and vegetables per day, low level of physical activity, overweight/obesity and raised blood pressure was 9.2%, 18.2%, 86.4%, 52.3%, 56.2% and 44.4% respectively. On multivariate analysis strong predictors of overweight obesity were age ≤59 years (AOR: 3.21, CI 1.49–6.89), poor economic status (AOR: 1.764, CI: 1.120–2.779) and low physical activity (AOR: 2.132, CI: 1.350–3.370). Predictors of raised blood pressure were age ≥75 years (AOR: 2.04, CI: 1.01–4.11), unmarried and widow (AOR: 1.815, CI: 1.108–2.971), higher economic status (AOR: 1.752, CI: 1.119–2.740) and smoking (AOR: 4.109, CI: 1.737–9.718).

**Table 3. Determinant of metabolic risk factors bivariate analysis.**

| Variable | Categories | Overweight/ obesity (%) | Raised blood pressure | Variable | Categories | Overweight/ obesity (%) | Raised blood pressure (%) |
|---|---|---|---|---|---|---|---|
| **Age (in years)** | ≤59 | 66.2 | 35.8 | **Smoking** | Yes | 57.6 | 22.2 |
| | 60–74 | 51.7 | 46 | | No | 41.7 | 46.6 |
| | ≥75 | 45.5 | 59.1 | | p—value | 0.066 | 0.005* |
| | p—value | 0.005* | 0.006* | **Alcohol** | Yes | 64.8 | 49.3 |
| **Marital status** | Married | 58.2 | 38.7 | | No | 54.2 | 43.3 |
| | Unmarried/ Widow | 52.2 | 55.2 | | p—value | 0.105 | 0.355 |
| | p—value | 0.26 | 0.002* | **Fruits and vegetables (servings)** | <5 | 55.8 | 45.4 |
| **Ethnicity** | Dalit | 59.3 | 55.6 | | ≥5 | 58.5 | 37.7 |
| | Janjati | 58.3 | 45.8 | | p—value | 0.712 | 0.296 |
| | Brahmin | 52.3 | 43.1 | **Salt consumption** | ≤5 gram | 56.3 | 45.7 |
| | Chhetri | 59.8 | 44.3 | | >5 gram | 56 | 41.8 |
| | Madhesi and Giri | 30.8 | 15.4 | | p—value | 0.958 | 0.46 |
| | p—value | 0.291 | 0.2 | **physical activity** | Low | 61.3 | 51.5 |
| **Education level** | Illiterate | 55.5 | 45.1 | | Normal | 50.5 | 44.4 |
| | Literate | 61.9 | 38.1 | | p—value | 0.033* | 0.003* |
| | p—value | 0.427 | 0.387 | **Family member** | | | |
| **Economic status** | Below Poverty Line | 52.6 | 39.9 | | <5 | 58.2 | 46.4 |
| | Above Poverty Line | 62.8 | 52.6 | | ≥5 | 54.9 | 43 |
| | p—value | 0.053 | 0.017* | | p—value | 0.519 | 0.513 |

*- significant

**Table 4. Multivariate analysis.**

| Variables | Categories | Overweight/obesity | Variables | Categories | Raised blood pressure |
|---|---|---|---|---|---|
| **Age in years** | ≤59 | 3.213 (1.497–6.896)* | **Age in years** | ≤59 | Ref |
| | 60–74 | 1.409 (0.707–2.81) | | 60–74 | 1.536 (0.95–2.483) |
| | ≥75 | Ref | | ≥75 | 2.035 (1.009–4.106)* |
| **Religion** | Hindu | Ref | **Marital status** | Married | Ref |
| | Others | 2.152 (0.696–6.652) | | Unmarried /widow | 1.815 (1.108–2.971)* |
| **Occupation** | Homemaker | 2.116 (0.996–4.494) | **Economic status (Poverty Line)** | Below | Ref |
| | Not working | 1.734 (4.494–4.526)* | | Above | 1.751 (1.119–2.74)* |
| | Self-employed | Ref | **Smoking** | No | Ref |
| **Economic status (Poverty Line)** | Below | Ref | | Yes | 4.109 (1.737–9.718)* |
| | Above | 1.764 (1.12–2.779)* | **Physical activity** | Normal | Ref |
| **Smoking** | No | Ref | | Low | 1.434 (0.92–2.236) |
| | Yes | 0.499 (0.236–1.057) | **Overweight/ obesity** | No | Ref |
| **Physical activity** | Normal | Ref | | Yes | 1.518 (0.977–2.359) |
| | Low | 2.132 (1.35–3.37)* | | | |

Ref- Reference group

*- Significant

In the current study, the prevalence of current smoking is found to be 9.2% and it is slightly higher than the 2019 NCD risk factors survey of Nepal that is 7.5% among women of age group 15–69 years [15] and the study conducted in United states among postmenopausal women (7%) [23]. Majority of the respondents (80%) started smoking before 15 years of age which is consistent with the findings of study conducted in Sindhuli district of Nepal (64.6%) [24]. In regards to current smokeless tobacco use, the prevalence in this study is observed to be 11.3% which is consistent with NDHS survey 2016 i.e.,10.8% among women of province one. However, higher than the 2019 NCD risk factors survey of Nepal that is 4.9% among women of age group 15–69 years [15].

The prevalence of alcohol consumption in past one month is 18.2%. This findings is slightly less than the 2019 NCD risk factors survey of Nepal among all adults of age group 15–69 years (20.8%) [15]. Lifetime abstainers was 79.7% in this study which is less than NCDs survey 2019 in Nepal (86.5%) among women of age group 15–69 years [15]. Similarly less than 5 servings of fruits and vegetables in this study was extremely high 86.4%. This finding is less than nation-wide NCDs survey in Nepal (96.3%) [15], India (98.4%) [25] and study conducted in Pakistan (96.5%) [26]. However, higher than the study conducted in Delhi (64.2%) [22]. Prevalence of salt consumption more than 5 gram per day in this study was 34.4%. Where as nearly half of the rural postmenopausal women of Bangladesh were found to consume more than 5 gram of salt per day [10].

More than half (52.3%) of respondents has been reported to have low physical activity, which is in the agreement with the study conducted among rural postmenopausal women in India (55%) [27] and postmenopausal women of the Bangladesh (58.1%) [10]. The prevalence is low in comparison to the study conducted in Brazil among climacteric women (87.2) [28].

High prevalence of Overweight/obesity being observed (56.2%) in this study. Finding of this study is higher in comparison to the study conducted in India (44%) [29] and United State (46%) [23]. Prevalence of overweight in this study is 31.5% which is less than study conducted in West Bengal (50%) [30] and Brazil (38.5%) [28]. However prevalence of overweight in this study was less than study conducted among postmenopausal women of rural India (78%) [21]. Prevalence of obesity in this study was 24.8% which is higher than study conducted in West Bengal (17%) [30] and Delhi (12.7%) [22]. The findings of this study reveal a strong association between overweight/obesity and age, economic status and physical activity on multivariate regression analysis. it is observed that the odds of being overweight/obesity increases by 3.21 times in age group ≤59 years and by 1.4 times among age group 60–74 years than age group ≥75 years. Prevalence of overweight/obesity decreases with increasing age after 60 years and highest among respondent younger than 59 years of age which is consistent with other study conducted in Kathmandu [14, 31], Spain [32] and Turkey [33]. Another significant factor associated with overweight/obesity is economic status. Respondents above poverty line were 1.76 times more likely to be overweight/obesity than those living below poverty. Similar with study conducted in Turkey [33].

This study has showed the prevalence of raised blood pressure 44.4% among post-menopausal women of the Itahari sub-metropolitan city of Nepal which is similar to the study conducted in Banepa where prevalence of raised blood pressure in older women was 42.73% [34] and Lamjung district of Nepal between aged 40–80 years(42.9%) [35]. This is less than case control study conducted in India among postmenopausal women (52%) [7] and study conducted in West Bengal which found prevalence of hypertension 64% among urban women [30]. Our study reported that age, marital status and economic status are strong predictors of raised blood pressure. This study revealed risk of raised blood pressure increases with advancing age and the risk is almost 2 times higher in age group 60–74 years and 2.04 times in age group ≥75 years than age group ≤ 59 years. Supported by study conducted in Kathmandu

[14, 36], Banepa [34], Eastern Ethopia [37] and China [38]. Odds of having raised blood pressure was 1.82 times higher among unmarried and widow women than the married women in present study. Study conducted in Iran reported that unmarried women or widow were at low risk for raised blood pressure in comparison to married women which is contrast to our study finding [39]. Similarly, it was observed that respondents belonging to above poverty line were 1.75 times more likely to have raised blood pressure than respondents belonging to below poverty line which was supported by study conducted in Dolakha district of Nepal [40]. However, this finding is contrast to the study conducted in China where income level was protective factors of hypertension [38]. Furthermore, smokers were at higher risk to have raised blood pressure than nonsmokers. This finding is supported by study conducted in China [38], rural Uttarkhanda [41], Kathmandu [36, 42] and Ethopia [37].

In this study 13.6% respondents had tested lipid profile within one month at the time of a data collection. Level of cholesterol and triglyceride was alarmingly high among respondents who have tested i.e., 49% had raised cholesterol level and 58.5% had raised triglyceride level in this study. Prevalence of raised cholesterol reported in this study is consistent with study conducted in India (52%) [29]. However, higher than the study conducted in rural India (30%) [21], Bangladesh (25.7%) [10] and review article of Latin America (21.6%) [43]. Likewise, 58.5% respondents had raised triglyceride level in this study which is higher than study conducted in rural India (31%) [21] and northern Bangladesh (44%) [44].

## Limitations

This study cannot be generalized to rural postmenopausal women as study was conducted in urban setting. There was a risk of recall bias while recording dietary history and assessing seven days physical activities.

## Conclusion

This study showed high prevalence of behavioural and metabolic risk factors of cardiovascular diseases especially low physical activity, inadequate vegetables and fruits intake, alcohol consumption, overweight and raised blood pressure in postmenopausal women of Itahari submetropolitan city of Nepal. This calls for an early need of interventions and policies at local, provincial and institutional level.

## Acknowledgments

We express heartfelt gratitude to School of Public Health and Community Medicine B.P Koirala Institute of Health Sciences, Dharan for granting us permission to conduct study on this topic. We are heartly grateful to Itahari sub-metropolitian city for permission to conduct study on Itahari. We would also likes to thanks every respondents and individuals who had contributed for this study.

## Author Contributions

**Conceptualization:** Suraksha Khatri, Deepak Kumar Yadav, Birendra Kumar Yadav, Paras Kumar Pokharel.

**Data curation:** Suraksha Khatri, Dharanidhar Baral.

**Formal analysis:** Suraksha Khatri, Dharanidhar Baral.

**Funding acquisition:** Suraksha Khatri.

**Investigation:** Suraksha Khatri, Deepak Kumar Yadav.

**Methodology:** Suraksha Khatri, Deepak Kumar Yadav, Dharanidhar Baral.

**Project administration:** Suraksha Khatri.

**Resources:** Suraksha Khatri.

**Software:** Suraksha Khatri, Dharanidhar Baral.

**Supervision:** Suraksha Khatri, Deepak Kumar Yadav, Anup Ghimire, Dharanidhar Baral, Birendra Kumar Yadav, Paras Kumar Pokharel.

**Validation:** Suraksha Khatri, Dharanidhar Baral.

**Visualization:** Suraksha Khatri.

**Writing – original draft:** Suraksha Khatri, Anup Ghimire, Birendra Kumar Yadav.

**Writing – review & editing:** Suraksha Khatri, Deepak Kumar Yadav, Dharanidhar Baral.

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
