## [Decision Letter · Decision Letter 0]

30 Oct 2023

PONE-D-23-21065“Study on Behavioural and Metabolic risk Factors of Cardiovascular Disease among Post-Menopausal Women of the Itahari Sub-Metropolitan city of Nepal”PLOS ONE

Dear Dr. Khatri,

Thank you for submitting your manuscript to PLOS ONE. After careful consideration, we feel that it has merit but does not fully meet PLOS ONE’s publication criteria as it currently stands. Therefore, we invite you to submit a revised version of the manuscript that addresses the points raised during the review process.

We look forward to receiving your revised manuscript.

Kind regards,

Dhan Bahadur Shrestha, MBBS

Academic Editor

PLOS ONE

Journal Requirements:

2. Thank you for stating the following financial disclosure: "Unfunded"

3. Thank you for stating the following in your Competing Interests section:  "No"

6. Please amend the manuscript submission data (via Edit Submission) to include authors: 

- Deepak Kumar Yadav

- Anup Ghimire

- Dharanidhar Baral

- Birendra Kumar Yadav

- Paras Kumar Pokharel

8. We note you have included a table to which you do not refer in the text of your manuscript. Please ensure that you refer to Table 3 in your text; if accepted, production will need this reference to link the reader to the Table.

**Additional Editor Comments:**

Reviewers have raised concern on methods particularly sampling and data analysis; please carefully consider those and address carefully and transparently. Additionally, write up of the manuscript is not aligned per the guideline, please consider guideline while revising the manuscript and proof read to avoid any language or grammar related issues or use assistance from native language speaker! 

Reviewers' comments:

Reviewer's Responses to Questions

**Comments to the Author**

1. Is the manuscript technically sound, and do the data support the conclusions?

Reviewer #1: Yes

Reviewer #2: Yes

Reviewer #3: Yes

2. Has the statistical analysis been performed appropriately and rigorously? 

Reviewer #1: Yes

Reviewer #2: Yes

Reviewer #3: Yes

3. Have the authors made all data underlying the findings in their manuscript fully available?

Reviewer #1: Yes

Reviewer #2: Yes

Reviewer #3: Yes

4. Is the manuscript presented in an intelligible fashion and written in standard English?

Reviewer #1: Yes

Reviewer #2: No

Reviewer #3: No

5. Review Comments to the Author

Reviewer #1: GENERAL:

1. Please write the full form of abbreviation for the first time in the text.

2. Please spell the single digit in the text (eg. Three for 3)

INTRODUCTION:

1. The fourth paragraph looks too lengthy and redundant. Please make it shorter and concise by including the main findings of different studies; rather than describing individual studies separately

METHODS:

1. The formula for sample size calculation is n = Z2pq/e2; not I2 (where e2 is the margin of error). Please correct this typo (probably) in your text.

2. The sampling method that you have described does not look like simple random sampling. Instead, it sounds like convenience sampling. Please clarify this aspect properly.

3. Please make sure that the font type, size and punctuations are appropriate in this section (check the authors guideline: https://journals.plos.org/plosone/s/submission-guidelines).

4. You have mentioned that “Explanatory variables showing p value less than 0.2 in bivariate analysis were entered into logistic model for multivariate analysis.” Could you please explain why this was done? Why was 0.2 taken as a level of significance at this point, and 0.05 while doing logistic regression later??

RESULTS:

1. Please mention below each table how p-value was obtained and its level of significance (eg. Bivariate analysis, logistic regresstion)

DISCUSSION:

1. Please mention the possible limitations of your study, and measures taken to overcome those limitations

Reviewer #2: Dear authors,

I would like to congratulate to the authors for being able to bring the manuscript to this level.

The manuscript was prepared based on a community-based cross-sectional study in Itahari, Nepal. Study population were post-menopausal women. Study found that prevalence of behavioral and metabolic risk factors is high among post-menopausal women of the study area. The study also determined the determinants of obesity/overweight and hypertension. The manuscript is well-prepared; however, I would suggest the following minor revisions to improve the manuscript.

Abstract:

• Authors have used STEP III as well (e.g lipid profile).

• Sampling technique is wrong. Authors have used multistage cluster random sampling, not simple random sampling.

Introduction:

• In fourth paragraph, study from Bangladesh came suddenly. Please either use general findings or describe studies from Nepal.

Method:

• Please change the name of methodology to method.

• This was not simple random sampling technique. This is the multistage cluster sampling. Therefore, the sample size calculation must reflect the design effect.

• Please change inadequate intake of fruits and vegetable intake instead of “Low fruits and vegetable intake”.

• One entire section is missing. How the outcomes were measured (e.g BP, salt intake, BMI etc.). Data is missing on how many times the BP was measured.

• Method does not describe about how data was collected for lipid profile. There is no information about blood sample intake and biochemical measurement. I would recommend to describe the data collection techniques properly, to be able to replicate the findings.

• I do not understand why authors have missed data of blood sugar, diabetes mellitus.

Result:

• What is the rationale for determining the determinants of obesity/overweight and raised blood pressure only?

• Could you provide the data on age of menopause. Early menopause is one of the risk factor of CVD (https://doi.org/10.1016/S2468-2667(19)30155-0 )

Discussion:

• Please write a brief summary of the key findings of the study at first paragraph.

• In some of the paragraphs, authors have missed the citation (e.g This finding highlights the need to regulate price of fruits and vegetables by government sectors and also to develop job opportunity to rise financial status of people (?), low consumption of fruits and vegetables among the respondents and also changing lifestyle because of urbanization and lack of availability of organic fruits and vegetables in the market (?).

• Discussion section is too long.

• Please add a paragraph on the implications for policy and future research recommendation.

• What are the limitations of the study?

Language edition

• Language needs to be corrected (e.g community-based CVDs risk factors prevention and management among postmenopausal women is the need of the day (?)

Reviewer #3: Throughout the manuscript, there are missing punctuations, make fonts and sizes uniform, and needs grammatical refinement.

Title: "risk" must start with capitalize letter.

Introduction: In any articles, fullform of words need to be mentioned prior using the abbreviations, need references for first and second lines in 5th paragraph.

In Data management and statistical analysis section in methods, "interquartile Range (IQR)" i in interquartile needs to be capitalized.

In socio-demographic characteristics of the participants in results, "majority (95.1%) of them were from Hindu religion" needs to be restructured to make it more coherent and fluent.

Behavioural and metabolic risk factors of CVDs in results: "Majority of the respondents were lifetime abstainer(79.7%). Current alcohol users 18.2%." bit confusing and need to be rephrased in a single sentence. In "Majority of the respondents 86.4%" write percent insides brackets. Need to correct spelling of "respiondents" in Metabolic risk factors.

Please mention the P-vlaues in AOR with 95% CI in "Determinant of metabolic risk factor". Mention statistically significant correlations at first and then non-significant ones.

In discussion: Instead of describing the prevalence of behavioural risk factors individually, it is better to write them together and correlate with their prevalence among cardiovascular disease patients, not with NCD risk factors survey findings as NCD is a umbrella term. Highlight more on the risk factors that are correlated significantly.

It is always good to point out some limitations that your study failed to address.

6. PLOS authors have the option to publish the peer review history of their article (what does this mean?). If published, this will include your full peer review and any attached files.

Reviewer #1: No

Reviewer #2: No

Reviewer #3: No

---

## [Author Response · Author response to Decision Letter 0]

20 Feb 2024

Correction has been made in accordance with the feedback of all reviewer.

---

## [Decision Letter · Decision Letter 1]

5 Aug 2024

PONE-D-23-21065R1“Study on Behavioural and Metabolic Risk Factors of Cardiovascular Disease among Post-Menopausal Women of the Itahari Sub-Metropolitan city of Nepal”PLOS ONE

Dear Dr. Khatri,

Thank you for submitting your manuscript to PLOS ONE. After careful consideration, we feel that it has merit but does not fully meet PLOS ONE’s publication criteria as it currently stands. Therefore, we invite you to submit a revised version of the manuscript that addresses the points raised during the review process. As pointed out by reviewers, some methods and write up needs more clarity and improvement. 

We look forward to receiving your revised manuscript.

Kind regards,

Dhan Bahadur Shrestha, MBBS

Academic Editor

PLOS ONE

Journal Requirements:

Reviewers' comments:

Reviewer's Responses to Questions

**Comments to the Author**

1. If the authors have adequately addressed your comments raised in a previous round of review and you feel that this manuscript is now acceptable for publication, you may indicate that here to bypass the “Comments to the Author” section, enter your conflict of interest statement in the “Confidential to Editor” section, and submit your "Accept" recommendation.

Reviewer #1: (No Response)

Reviewer #2: (No Response)

2. Is the manuscript technically sound, and do the data support the conclusions?

Reviewer #1: Yes

Reviewer #2: Yes

3. Has the statistical analysis been performed appropriately and rigorously? 

Reviewer #1: I Don't Know

Reviewer #2: Yes

4. Have the authors made all data underlying the findings in their manuscript fully available?

Reviewer #1: Yes

Reviewer #2: Yes

5. Is the manuscript presented in an intelligible fashion and written in standard English?

Reviewer #1: Yes

Reviewer #2: No

6. Review Comments to the Author

Reviewer #1: Dear authors, thank you for sending the revised version of the manuscript. But, still, I have following comments to be addressed from your side:

1. First of all, I would like to strongly suggest you to adhere to the submission guidelines of PLOS ONE journal (https://journals.plos.org/plosone/s/revising-your-manuscript). While submitting your review, you are required to reply to every comment one by one; not just replying in one line. You need to submit both marked-up copy and a clean copy of your manuscript while submitting revisions.

2. It is seen that you have not addressed the previous concerns regarding statistics part of your study. You have mentioned that while performing bivariate analysis, a significance level at p=0.2 was taken, whereas it was taken as less than 0.05 while doing logistic regression. Could you please explain it why it was done like this? Are you aware that this is statistically correct?

Reviewer #2: Dear authors,

Congratulations on submitting the revised manuscript. The current version is much improved for content and clarity. Yet, there is room for improvement.

Positive comments

The study focused on a group of population who are prone to have CVD. Data collection, analysis and interpretation are robust. The drawn conclusions are based on the presented data.

Comments to improve the manuscript.

Title

The title can be improved to truly represent the study. The word cardiovascular disease is cardiovascular diseases. I suggest to change the title as follows or something similar.

“Behavioural and Metabolic Risk Factors of Cardiovascular Diseases among

Post-Menopausal Women: A Cross-sectional Study in Itahari Sub-Metropolitan city of Nepal”

Introduction

In paragraph 3, the citation for the reported data is missing and the use of quotation is not a proper in this context. This comment also applies to other sentences. Authors are requested to provide the citation for the results, data, and conclusions beyond the current study.

Both men and women are living longer. The sentence (para 4, second sentence) is misleading.

The last sentence related to Itahari is to be written in the setting of the methodology.

Methods

I reiterate changing the methodology to the methods. The described sampling techniques suggest that it is a multistage cluster sampling. Authors may refer the reference 18 for confirmation.

The design effect for cluster sampling is to be adjusted in the sample size calculation. The data within the cluster will be similar and correlated. This effect of intra-cluster correlation can be minimized by inflating the sample size. The sample size derived from the provided formula needs to be increased by multiplying it by the design effect. These issues can be stated in the method section and discussed in the limitation.

How the study handled eligible participants not available during the data collection? These participants may have been hospitalized for CVD or related risk factors. This may underestimate the burden of risk factors. Authors can discuss the methodological limitations in the discussion section before reaching on conclusions.

In behavioural and metabolic risk factors, the separate headings are redundant. Meaning is clear even without those subheadings. The same is true in the result section as well.

How has economic status been defined?

How was data of lipid profile obtained? Did the study analyse blood sample?

The sentence “P-value less than 0.05 at 95% class interval(CI) was considered statistically significant” is not clear. The P-value to be changed to p-value or the sentence could be rephrased.

The consent was obtained from IRC or from participants?

Results

The footnotes of each table can be written in the last row.

Table 4: Please make the reference category consistent. What are the justifications for different reference categories for age in years to predict its effect on overweight and blood pressure?

What variables were adjusted in estimating the AOR?

Discussion

The limitation goes above the conclusion. This should be in a paragraph, not in bullet points.

General comments that apply to all sections.

1. Please provide the line numbers that will make it easier to refer to the comments.

2. By aligning the text to the left, spotting the extra space will be easier.

3. Language is to be corrected further. The glaring error is the improper use of the article. For example, “menopause is defined as the cessation of menstruation for 12 months in a woman over age of 45 years and occurs at a median age of 52 years. Correction to “over the age of 45 years” will be correct. Others are subject verb agreements (SVA). This sentence “Usually women experiences first episode of CVD after menopause and this period is landmark for impending CVDs” has inaccurate use of SVA.

4. The non communicable diseases (NCDs) is to be changed to noncommunicable diseases(NCDs) or non-communicable diseases(NCDs).

5. The facts are mostly reported in the present perfect tense (e.g authors can reconsider: the introduction, paragraph 3, last sentence: please check the sentence “that” is missing before total)

6. Done is not the proper word to refer to the conducted or carried out.

7. Use of the word postmenopausal, post menopausal, post-menopausal women are inconsistent.

8. The use of comparative is wrongly presented. One example is “If more than one post-menopausal women present at the time of data collection than one respondent was selected by using lottery method. If no postmenopausal women available than proceeded to adjacent household”.

9. Spelling error: respiondents

7. PLOS authors have the option to publish the peer review history of their article (what does this mean?). If published, this will include your full peer review and any attached files.

Reviewer #1: No

Reviewer #2: No

---

## [Author Response · Author response to Decision Letter 1]

14 Aug 2024

Good morning, changes based on reviewers feedback has been done and file is uploaded.

Kind regards

---

## [Decision Letter · Decision Letter 2]

6 Sep 2024

“Behavioural and Metabolic Risk Factors of Cardiovascular Diseases among

Post-Menopausal Women: A Cross-sectional Study in Itahari Sub-Metropolitan city of Nepal”

PONE-D-23-21065R2

Dear Dr. Khatri,

We’re pleased to inform you that your manuscript has been judged scientifically suitable for publication and will be formally accepted for publication once it meets all outstanding technical requirements.

Kind regards,

Dhan Bahadur Shrestha, MBBS

Academic Editor

PLOS ONE

Additional Editor Comments (optional):

Reviewers' comments:

Reviewer's Responses to Questions

**Comments to the Author**

1. If the authors have adequately addressed your comments raised in a previous round of review and you feel that this manuscript is now acceptable for publication, you may indicate that here to bypass the “Comments to the Author” section, enter your conflict of interest statement in the “Confidential to Editor” section, and submit your "Accept" recommendation.

Reviewer #1: (No Response)

Reviewer #2: All comments have been addressed

2. Is the manuscript technically sound, and do the data support the conclusions?

Reviewer #1: Yes

Reviewer #2: Yes

3. Has the statistical analysis been performed appropriately and rigorously? 

Reviewer #1: I Don't Know

Reviewer #2: Yes

4. Have the authors made all data underlying the findings in their manuscript fully available?

Reviewer #1: Yes

Reviewer #2: Yes

5. Is the manuscript presented in an intelligible fashion and written in standard English?

Reviewer #1: Yes

Reviewer #2: Yes

6. Review Comments to the Author

Reviewer #1: Dear Authors,

I am still not fully satisfied with your response regarding using different levels of significance in bivariate and multivariate analyses.

Taking significance level of 0.2 is a very large value compared to margin of error = 10% (as taken in sample size calculation); in this situation you can take up to 0.1 only as per my knowledge

Reviewer #2: Dear authors, thank you for addressing the previous comments.

This paper could be accepted after correcting the language.

Kind regards,

7. PLOS authors have the option to publish the peer review history of their article (what does this mean?). If published, this will include your full peer review and any attached files.

Reviewer #1: No

Reviewer #2: No

---

## [Editor Report · Acceptance letter]

13 Sep 2024

PONE-D-23-21065R2 

PLOS ONE

Dear Dr. Khatri, 

I'm pleased to inform you that your manuscript has been deemed suitable for publication in PLOS ONE. Congratulations! Your manuscript is now being handed over to our production team.

Kind regards, 

on behalf of

Dr. Dhan Bahadur Shrestha 

Academic Editor

PLOS ONE